# Parental attitudes to randomised controlled trials in primary dental care: A qualitative study

Heather Coventry[1]*, Anne Maguire[1], Elaine McColl[2], Catherine Haighton[3]

**1** Child Dental Health, School of Dental Sciences, Newcastle University, United Kingdom, **2** Population Health Sciences, Faculty of Medical Sciences, Newcastle University, United Kingdom, **3** School of Communities and Education, Faculty of Health and Wellbeing, Northumbria University, United Kingdom

* heather.coventry@ncl.ac.uk

## Abstract

### Background

A clinical paedodontic randomised controlled trial (FiCTION) provided the opportunity to explore recruitment and retention challenges in The National Health Service (NHS) primary dental care settings.

### Purpose

To investigate parental attitudes towards their child's participation in a dental randomised controlled trial (RCT).

### Methods

Parents whose child(ren) (aged 3–7 years) were participants in a dental RCT or who had been screened for the trial but did not participate were asked to consent to be contacted regarding completing a questionnaire and a semi-structured face-to-face qualitative interview. Using a purposive, maximum variation sampling strategy a sub-sample of parents who had completed the questionnaire study completed an interview. Data were coded using NVivo and the Framework Method of thematic analysis applied.

### Results

The 18 parents consenting to an interview indicated positive attitudes towards research in primary dental care. There were no noticeable contrasting views of good dental health, or perceptions of facilitators and barriers thereof, between parents whose child(ren) were FiCTION participants and those whose child(ren) were not. Research involvement did not appear to be a major incentive to attend a particular practice, and while parents viewed research-active dental practices favourably, they did not always understand why their practice was research-active (especially for those not participating in FiCTION). Most FiCTION parents felt comfortable with

**Data availability statement:** All relevant data are within the paper. The submission contains all raw data to replicate the results of the study.

**Funding:** The research was funded by the Centre of Oral Health Research at Newcastle University and NHS Education for Scotland (through the Scottish Dental Practice Based Research Network), awarded to HC. The funders did not play any role in the study design, data collection and analysis, decision to publish, or preparation of the manuscript.

**Competing interests:** The authors have declared that no competing interests exist.

the concept of trial withdrawal or requesting a change in treatment arm. However, parents did not always have complete knowledge or understanding of the research study in which they had been invited to participate. While FiCTION parents had overall greater understanding of research, concepts such as randomisation were hard for most parents to grasp.

## Conclusions

Parents valued dental research in primary care but perceived it as complex and challenging. Further research should explore the best methods to achieve engagement with patients in primary dental care research.

## Introduction

Despite the vast majority of dental services being provided in primary dental care, few randomised controlled trials (RCTs) have taken place or are underway in this setting [1]. The National Institute for Health and Care Research (NIHR) Health Technology Assessment funded FiCTION Trial – Filling Children's Teeth: Indicated or Not? [2] was a UK-based, multi-centre, 3-arm parallel group, patient-randomised trial to compare three treatment strategies used to manage decay in primary teeth [3]. Many RCTs report challenges with recruitment [4–6] and this was echoed with the FiCTION RCT [7] which aimed to recruit 1461 children aged 3–7 years [2]. A general reluctance, particularly among parents and healthcare professionals, to involve children in trials, has exacerbated the challenges in recruiting children to research studies [8]. Parents and healthcare professionals are potential gatekeepers who can facilitate or obstruct child participation in research, and it is important to consider the impact on both groups when a child has been considered for inclusion in research [9]. Published systematic reviews identifying problems with recruitment and retention of child patients to clinical trials have not primarily focused on parents [10]. Parent-related barriers and facilitators to research involvement have been considered in a limited number of research studies [11], of varying qualities and designs, but none have been within the dental field. More generally, it is commonly suggested that ethnic minorities, as well as lower income, poorly educated or lower socioeconomic status groups are less likely to participate in research and therefore are under-represented [12].

Learning from parents who decline their child from participating in dental research is critical if we aim to have a more representative research population and subsequently a more valid and generalisable evidence base. Dental research has shown that children living in the most deprived areas of the country are more than twice as likely to have experienced dentinal decay (32.2%) as those living in the least deprived areas (13.6%) [13]. There is also evidence of ethnic inequalities in oral health within the UK, largely based on limited data from England [14]. Lower participation of under-represented groups threatens to exacerbate existing paediatric dental health disparities. Understanding potential barriers related to parents not fully

understanding the importance of dental research in the context of paediatric dental care needs to be more robust to help shape policies to address these inequalities.

There is growing recognition that dental practices in The National Health Service (NHS) primary care provide an excellent environment in which to carry out clinical dental research [15]. However, trials in this environment, particularly those involving children remain rare, with limited evidence upon which to base recommendations to optimise participation in such trials [7]. However, it can be postulated that a parent's willingness for their child to be screened, randomised and retained in a dental RCT until completion could be influenced to some extent by their previous experiences of research, trust in the research being undertaken, as well as the behaviour and attitude of dental teams. For example, parents with previous research experience may have greater understanding that, whilst they had no control over what intervention their child was to receive, they could, at any stage, ask their dentist to withdraw their child from the study, making them less worried about study enrolment. Conversely, parents with no prior research participation experience may fear potential harm to their child, such as short- or long-term adverse effects, resulting in reluctance to consent to their child's participation.

When the FiCTION RCT was funded, it was important to fully understand any initial or emerging recruitment or retention challenges in this setting. It was similarly unclear whether parents' perception of their own dental health might act as a facilitator or a barrier to their child participating in dental research. Recruitment remains an important area to address, if dental research is to flourish and address key dental health research questions. Therefore, when the opportunity arose during the FiCTION trial, this qualitative study was undertaken to investigate parents' views, knowledge and experience around their child's involvement in dental research, based on their participation (or not) in the FiCTION RCT. Specifically we wanted to explore parents' views, knowledge and experience regarding: their own dental health and their families' dental care; their participation in research as a participant or participant's parent; any contrasting views therein between those parents whose children were participating in the FiCTION RCT and those parents whose children were not participating in that RCT, either because they were ineligible or because the parent had declined consent.

## Study objectives

The objectives associated with the qualitative study were:

1. To investigate parents' views, knowledge and experience regarding their own dental health and their families' dental care and any differences therein between those parents whose children were participating in the FiCTION RCT and those parents whose children were not participating in that RCT.

2. To investigate parents' views, knowledge and experience about participation in research and any differences between parents whose children were participating in the FiCTION RCT and those whose children were not participating in that RCT.

## Methods

**Design.** The research project, conducted in the context of a PhD [16] comprised two standalone components: a longitudinal questionnaire-based survey (at baseline and 18 months) of parents whose children were FiCTION RCT participants or had been screened for FiCTION but did not (or could not) participate; semi-structured face-to-face qualitative interviews with a subsample of these parents. This project was nested alongside the UK-based FiCTION Trial and was conducted within two of the five geographical centres for FiCTION (Scotland and North-East England) in those practices where general dental practitioners had additionally agreed to participate in this study. The choice of these two centres was to allow differences in geographical location (urban versus rural), dental service funding and community-based oral health prevention programmes to be considered. It was felt that incorporating all five centres would result in

increased complexity of data collection, with limited additional benefit and would not be cost efficient. The qualitative study is reported here, based on published standards for reporting qualitative research recommendations [17].

Of the 42 dental practices contacted, 13/26 (50%) FiCTION practices in Scotland and 14/16 (88%) FiCTION practices in north-east England agreed to take part in the research project. In total, across all 27 dental practices, 2980 parents, slightly fewer than expected, were identified as being potentially eligible for the research project and subsequently sent invitation packs and asked to return a completed consent form, they were only included if they opted in to the research project. In total, 332 parents (11.1%) of those sent the research project invitation pack returned the consent form. Of these parents, the child's FiCTION status was ascertained as; Child participating in FiCTION (n = 66), Child not participating in FiCTION as family declined (n = 6) and Child not participating in FiCTION as not eligible (n = 258). As planned, the six parents whose children were not participating in FiCTION, as the family had declined, were combined with the parents whose children were not participating in FiCTION, because they were not eligible, to produce a single Non-FiCTION group (n = 264).

Almost all of the 332 parents (n = 312, 94.0% of those returning a consent form, 10.5% of those contacted) who returned the consent form agreed to participate in the research project. The rate of consent was very similar irrespective of the child's FiCTION status. All consenting parents met the research project criteria and were then sent a baseline questionnaire. In total, and following the issue of reminder questionnaires to those who did not respond to the initial mailing, 261 parents (84% of those consenting to the research project) returned a completed baseline questionnaire and were included in the study (Table 1). It can be seen, by comparing the country of residence, that more parents were recruited from England than Scotland, irrespective of FiCTION status (overall 65.1% England, 34.9% Scotland); however, this was proportional to the number of invitation packs sent in England and Scotland. There was the same ratio of participants from England and Scotland in both the FiCTION and Non-FiCTION status groups.

The baseline study sample was 261 parents residing in the UK; 55 FiCTION parents and 206 Non-FiCTION parents. The mean age of parents was 38.1 years (SD 5.8 years), and 54% (140/261) were in the age group 36–45 years. There was little baseline difference between parents in the FiCTION and Non-FiCTION groups in terms of their age, relationship to the child or ethnicity (Table 2). The majority of questionnaires, irrespective of FiCTION status, were completed by mothers and by parents who identified themselves as white. There were large differences in the highest education level completed between the FiCTION and Non-FiCTION groups, with higher levels of educational attainment amongst the Non-FiCTION group (29.5% Non-FiCTION completed postgraduate education versus 16% FiCTION), suggesting either a different demographic mix of parents, a differing understanding of terminology used to describe education attainment or simply chance variation.

## Ethics

A favourable ethical opinion was obtained from the National Research Ethics Service (NRES) Committee North East – Newcastle and North Tyneside 1 (REC Reference: 13/NE/0180, Date: 26/11/2013). The project was conducted in accordance with the ethical principles set out in the Declaration of Helsinki (2013) [18]. Research and Development (R&D)

**Table 1. Number of questionnaires returned by FiCTION and Non-FiCTION parents at baseline and after 18 months, in Scotland and England.**

| | FiCTION | | Non-FiCTION | | Total | |
|---|---|---|---|---|---|---|
| | Baseline | 18 months from baseline | Baseline | 18 months from baseline | Baseline | 18 months from baseline |
| | n | n | n | n | n | n |
| Scotland | 19 | 16 | 72 | 58 | 91 | 74 |
| England | 36 | 29 | 134 | 97 | 170 | 126 |
| Total | 55 | 45 | 206 | 155 | 261 | 200 |

**Table 2. Demographic characteristics of FiCTION and Non-FiCTION parents at baseline.**

| Baseline characteristics | FiCTION (n = 55) | | Non-FiCTION (n = 206) | | Total (n = 261) | |
|---|---|---|---|---|---|---|
| | n | | n | | n | |
| **Age (years)** | | | | | | |
| Mean | | 38.1 | | 38.1 | | 38.1 |
| Standard Deviation | | 6.7 | | 5.5 | | 5.8 |
| Total responses | 47 | | 197 | | 244 | |
| | | | | | | |
| **Relationship to child** | | | | | | |
| Mother | 46 | (86.8%) | 182 | (88.3%) | 228 | (88.7%) |
| Father | 7 | (13.2%) | 16 | (7.8%) | 23 | (8.9%) |
| Other | 0 | (0.0%) | 6 | (2.9%) | 6 | (2.3%) |
| Total responses | 53 | | 204 | | 257 | |
| **Ethnicity** | | | | | | |
| White | 53 | (96.4%) | 197 | (96.1%) | 250 | (96.2%) |
| Indian, Pakistani or Bangladeshi | 0 | (0.0%) | 5 | (2.4%) | 5 | (1.9%) |
| Mixed race | 0 | (0.0%) | 3 | (1.5%) | 3 | (1.2%) |
| Other | 2 | (3.6%) | 0 | (0.0%) | 2 | (0.8%) |
| Total responses | 55 | | 205 | | 260 | |
| **Highest level education completed** | | | | | | |
| Primary school | 4 | (8.0%) | 5 | (2.5%) | 9 | (3.4%) |
| Secondary school | 12 | (24.0%) | 13 | (6.5%) | 25 | (9.6%) |
| Some additional training | 12 | (24.0%) | 63 | (31.5%) | 75 | (28.7%) |
| Undergraduate university | 14 | (28.0%) | 60 | (30.0%) | 74 | (28.4%) |
| Postgraduate university | 8 | (16.0%) | 59 | (29.5%) | 67 | (25.7%) |
| Total responses | 50 | | 200 | | 250 | |
| **Country of Residence** | | | | | | |
| England | 36 | (65.5%) | 134 | (65.0%) | 170 | (65.1%) |
| Scotland | 19 | (34.5%) | 72 | (35.0%) | 91 | (34.9%) |
| Total responses | 55 | | 206 | | 261 | |

management approval was obtained, in North-East England from the North of England Commissioning Support (NECS) and in Scotland from the NHS Research Scotland (NRS) Permissions Coordinating Centre. Parents provided written informed consent.

## Sampling and recruitment

The sampling method chosen was purposive [19] to ensure that the knowledge, views and experiences of chosen parents were explored in detail and that a full range of parents were included. For the purposes of the study, all study parents were required to have had their child screened for the FiCTION RCT as this was central to the research question. A number of relevant factors were also considered (Table 3) when designing the sampling strategy.

Maximum variation selection [20] of parents was used as it was felt that this enabled all the variables to be incorporated while maximising the diversity of data collected to address the research question. Parents who had returned the baseline questionnaire for the quantitative aspect of this study [16] and had signed the consent form agreeing to be contacted

**Table 3. Parameters used to select parents for the qualitative study.**

| Variable | Description |
|---|---|
| Participation in RCT (FiCTION) | Child participating; Child eligible but not participating; Child not eligible |
| Geographical location (UK) | Scotland; North-East of England |
| Ethnicity of parent | White; Other |
| Gender of parent | Female; Male |
| Age of parent | 24 years or under; 25–44 years; 45 + years |

regarding the qualitative study were eligible to participate; 55 FiCTION parents and 206 Non-FiCTION parents. A covering letter explaining the qualitative study, and including the Parent Information Sheet and Consent Form (for information purposes only), was sent to potential parents and a follow up telephone call made two weeks later by the interviewer (HC). Further written consent was obtained from all parents who agreed to participate prior to the interview commencing. In planning the study and applying for ethical approval, it was anticipated that data saturation, when further interviewing would generate no additional themes [20], would be reached at between 14 and 25 interviews; however, the actual number of interviews undertaken was determined by the achievement of data saturation.

## Qualitative approach and rationale

The purpose of each interview was to investigate, in depth, a parent's reality, their experiences and how they made sense of them. With the aim of having a 'conversation with a purpose' [21], open questioning was adopted throughout to explore each parent's experiences thoroughly [22]. To ensure that the specific areas of interest were studied, the conduct of interviews was guided by a pre-prepared interview topic guide (see supplement). As the aim of the study was to generate detailed and in-depth descriptions of human experiences, the approach taken included some of the characteristics associated with phenomenological interviewing [22]. For example, the interviewer (HC) took a neutral stance and refrained, as far as possible, from evaluating or challenging the parent's responses, to enable the interviewee to feel comfortable in providing in-depth descriptions of the areas of interest. However, as a semi-structured approach was used and each parent was only interviewed once, the qualitative study did not comply with all the principles associated with phenomenological interviewing. The topic guide was developed via discussion with the study team and in light of findings from the FiCTION pilot study [23], as well as another study [24] which had examined medical patients' understanding and knowledge with respect to participation in an oncology trial. Prior to commencing the interviewers, the lead author (HC) conducted pilot interviews with two volunteers: one familiar and one unfamiliar with the FiCTION RCT. As well as enabling testing of the practicalities associated with the study methods, the pilot also helped this author to develop her interviewing and transcribing techniques. Subsequent transcription and discussion with the other authors allowed the suitability of the topic guide to be examined. Based on the pilot work, the topic guide was slightly amended and re-organised to improve the natural flow of the conversation. No unexpected emerging topics were identified at this stage and the pilot data collected were then discarded. During the study interviews, topics relating to a parent's experience and views were explored before introduction of topics testing knowledge and comprehension of research, to help each interviewee feel more at ease. The final topic guide ensured that all parents were asked about the anticipated areas of interest, but it evolved further as interviews continued and additional related subject areas and emerging topics were identified (see supplement).

## Data collection

Each semi-structured interview was recorded using a digital recorder then transcribed verbatim and anonymised [25, 26]. The interviews were carried out at a convenient time and location for the interviewee; either in the interviewees' home, or

in a public place. Since interview location has been shown to be important in terms of power relations between researchers and research subjects [27, 28], a non-clinical environment was selected and the interviews were conducted in a comfortable venue with any costs to the parent minimised. Interviews were conducted between August 4th and October 26th 2015. No field notes were made during and/or after the interviews. The transcribed data were entered into NVivo Version 11 [29] and the transcript was checked against the original audio recordings to ensure accuracy. For quality control purposes, 300 words from two interviews were transcribed twice and the transcripts compared to ensure standards were being maintained. The digital recordings were stored on a Newcastle University password-protected PC and then deleted once transcriptions had been checked. To ensure that themes emerging from one interview were processed and used to inform the next interview in a consistent way, HC conducted all the interviews and all the transcription and analysis. By reviewing each interview during transcription, data entry, coding and afterwards, together with some secondary review with the study team, ways to improve the interview style were highlighted and acted upon.

The interviewer (HC) was employed as a dentist as well as a PhD student at the time of this study. Prior to this qualitative study, she had no experience of qualitative interviewing and therefore underwent training before commencing the study. None of the parents were known to the research team prior to the study commencing. The parents were told that the interviewer was interested in obtaining a fuller picture of their thoughts and experiences of being involved in research and how they felt about their dental health and that of their child(ren). The interviewer introduced herself as a PhD student and researcher. By introducing herself solely as a student and researcher, rather than as a qualified dentist, it was hoped that the interviewees would be more likely to talk freely without worrying about discussing any negative experiences in relation to their dental care and attitudes to dental health. However, in instances where she was asked directly, she disclosed that she was a dentist and accepted that this may have affected the data collected. It was acknowledged that HC's clinical background may have influenced her interpretation of the data being collected and analysed, both consciously and subconsciously. She therefore adopted the ontological approach of subtle realism by acknowledging that she may have impacted the research with subjective perceptions and understandings used in the interpretation of the data collected.

## Data analysis

The methodological orientation used to underpin this study was thematic analysis [30]. An essentialist method was used initially when exploring parents' dental experiences to focus on what they felt truly mattered to them. A constructionist method which supported parents to reflect and make sense of their lived experiences was then employed when discussing their decisions regarding their children's dental journey and entry into screening for the FiCTION RCT [30]. A contextualist method was then used with parents by the researchers to check their understanding of the information the parent had given, to enable further discussion and allow clarification. These methods are all used under the umbrella of thematic analysis. Thematic analysis was selected as it can both reflect reality (experiences, meanings and the reality participants have reported), and unpick the surface of "reality" (the ways individuals have made a meaning of their experiences) [30]. Thematic analysis has been suggested as suitable for those early in their qualitative research carer. NVivo [29] was used as a tool to manage the transcripts and to select and label sections of dialogue, thereby creating a method of indexing. Sections were highlighted and grouped together into electronic files. The data were then collated into potential themes which enabled HC, as the data analyst, to become familiar with the data collected. The Framework Method of qualitative analysis [31] was then applied to the dataset [32] as a method to organise and manage the dataset. At several points during the interviewing and transcribing processes, HC discussed potential themes with fellow authors. This, in combination with the development and application of the framework matrix to the data set, allowed theoretical concepts (both prior concepts and those emerging from the data) to be explored. All the data generated in the qualitative study were examined independently by the authors to ensure the themes generated were valid. Where there was disagreement, this primarily related to coding of the data, and this was resolved through discussion. Once analysis was finalised, parents were sent a summary of the themes that had been formed during the analysis of the complete data set. Parents were asked to contact

the interviewer if they felt the themes were not an accurate representation of the data. No parents contacted the interviewer with comment and/or correction. Key findings are reported under each theme using appropriate verbatim quotes to illustrate findings. Quotations from parents are used to strengthen each theme's credibility in terms of fairness and accuracy [33].

## Results

Nineteen parents were approached to take part in the research to provide a range of different views, at different locations within Scotland and North-East England. In total, data were collected from 18 of these parents; one parent was uncontactable as they did not answer their telephone on repeated occasions. A participant flow chart is given in Fig 1.

The lead author (HC) confirmed with each parent that they personally had been responsible for completing at least a baseline questionnaire in FiCTION. All parents had accompanied their child to their dental appointments. Data collection was stopped after 18 interviews at the point of 'informational redundancy' (the point at which the researcher began to hear the same comments and felt the new data was redundant of the data already collected) and it was felt that saturation had been reached after the 18th interview [34]. Characteristics of participating parents are given in Table 4.

The mean age of parents was 40.4 years (SD 7.3 years), which was slightly higher than the full baseline study sample (n = 261) where the mean age of parents was 38.1 years (SD 5.8 years). Most parents were interviewed at home (n = 16) and the others were interviewed in a coffee shop close to their home (n = 2). The interviews ranged in duration from 20 to 90 minutes.

### Dental health

Each parent's historical and current dental knowledge, views and experiences regarding their own dental health and their family's dental care was explored. In addition, the parent's dental history from childhood into adulthood was explored, along with the effect this may have had on their dental experiences and their decision to allow their child to participate in the FiCTION RCT or not (due to ineligibility for/unwillingness to take part in FiCTION). During the analysis of the parent's views, knowledge and experiences, three major themes and associated sub-themes were identified:

1) Good dental health is important; it means that dental issues have been dealt with, and individuals are educated about dental health;

2) Poor dental health impacts on nutritional, psychological and social performance, and;

3) Parents' dental practice selection and their dental attendance have nothing to do with research.

Below, each theme includes exploration of any contrasting views between FiCTION and Non-FiCTION parents.

**Theme 1: Good dental health is important; it means that dental issues have been dealt with and individuals are educated about dental health.** The findings from the two groups of parents (FiCTION and Non-FiCTION) were broadly similar. There were no noticeable contrasts between the two sets of parents in their opinions of good dental health or their perception of the facilitators of and barriers to achieving good dental health. Parents' views on the impact of poor dental health on nutritional, psychological and social performance were also very similar:

*"I know my mum and dad have got dentures now and they can't eat things like a lot of fruit and things like that, like sort of apples and things. They won't eat because of…they can't, basically." (Parent 7, Non-FiCTION (Ineligible), Female, 34 years old)*

*"You can't help but make assumptions if someone sort of got like the front teeth missing. It's like they've maybe had been in prison or that happens during a fight or you can't help do but...." (Parent 9, Non-FiCTION (Ineligible), Female, 33 years old)*

| Parents across 27 dental practices in Scotland and North-East England sent research invitation packs  n = [2980] |
|---|
| (Scotland n = 941, NE England n = 2039) |
| *Parent did not respond to invitation packs  n = [ 2648]* |

↓

| Parents responded to invitation pack and agreed to participate  n = [332] |
|---|
| FiCTION (n = 66), Non-FiCTION (n=264), Not screened for FiCTION (n=2) |
| *Parent responded to invitation pack but declined to participate  n = [ 20]* |

↓

| Parents consented to participate in two standalone components of research project  n = [312] |
|---|
| FiCTION (n = 64), Non-FiCTION (n=248) |

↓

| Parent returned completed baseline questionnaire  n = [261] |
|---|
| FiCTION eligible and joined (n = 55) (Scotland n = 19, NE England n = 36) |
| FiCTION eligible but declined (n=6) (Scotland n = 2, NE England n = 4) |
| Not eligible for FiCTION (n=200) (Scotland n = 70, NE England n = 130) |

↓

| Parents invited to participate in qualitative interview  n = [19] |
|---|
| FiCTION eligible and joined (n = 7) (Scotland n = 4, NE England n = 3) |
| FiCTION eligible but declined (n=2) (Scotland n = 1, NE England n = 1) |
| Not eligible for FiCTION (n=10) (Scotland n = 6, NE England n = 4) |

↓

| Parents completed qualitative interview  n = [18] |
|---|
| FiCTION eligible and joined (n = 7) (Scotland n = 4, NE England n = 3) |
| FiCTION eligible but declined (n=1) (NE England n = 1) |
| Not eligible for FiCTION (n=10) (Scotland n = 6, NE England n = 4) |
| *Parent withdrew consent to participate in qualitative interview  n = [1]* |

**Fig 1. Participant flow chart.**

*"If you do have poor dental health and you do have funny breath, you are not going to want that person to meet people and represent your company." (Parent 15, Non-FiCTION (Ineligible), Female, 35 years old)*

Only two parents, one from each group, identified that poor dental health could impact on an individuals' verbal and non-verbal communication:

*"It could affect your speech." (Parent 12, Non-FiCTION (Ineligible), Female, age not reported)*

**Table 4. Characteristics of parents participating in structured interviews.**

| Participation in FiCTION | Geographical location | Ethnicity of parent (White, Other) | Gender of parent | Age of parent | Highest level education completed | Parent ID |
|---|---|---|---|---|---|---|
| **Child participating in FiCTION (FICTION)** | North-east England | Other (Iraqi – Arab) | Male | Not reported | Not reported | 1 |
| | North-east England | White | Female | 32 | Undergraduate | 5 |
| | Scotland | White | Female | 34 | Undergraduate | 6 |
| | Scotland | White | Female | 42 | Undergraduate | 8 |
| | Scotland | White | Male | 51 | Postgraduate | 13 |
| | North-east England | White | Male | Not reported | Primary | 14 |
| | Scotland | Other (Nigerian -Yoruba) | Male | Not reported | Primary | 17 |
| Child not participating in FICTION as not eligible (Non-FiCTION) | North-east England | White | Female | 43 | Some additional training | 2 |
| | North-east England | Other (Indian) | Male | 41 | Postgraduate | 3 |
| | Scotland | White | Female | 34 | Undergraduate | 7 |
| | Scotland | White | Female | 33 | Postgraduate | 9 |
| | Scotland | Other (Mixed race – Pakistani/White) | Female | 44 | Some additional training | 10 |
| | Scotland | White | Male | 31 | Postgraduate | 11 |
| | Scotland | Other (Pakistani) | Female | Not reported | Undergraduate | 12 |
| | North-east England | White | Female | 35 | Secondary | 15 |
| | North-east England | White | Male | 43 | Some additional | 16 |
| | Scotland | White | Male | 56 | Undergraduate | 18 |
| **Child not participating in FiCTION as family declined (Non-FiCTION)** | North-east England | White | Female | 47 | Some additional training | 4 |

National guidelines for dental attendance recommend that patients whose dental disease activity continues unabated may need a shorter recall interval and closer supervision than those with stable oral health. If practices follow these recommendations, it would be anticipated that FiCTION-eligible children might attend dental practices more frequently. It was only Non-FiCTION parents who were not themselves on a regular maintenance programme. This was noteworthy as it was reported for both subgroups of Non-FiCTION parents, i.e., those where the child was eligible but not participating as the parent had declined, and where the child was not eligible. This could be due to the frequency of the FiCTION recall appointments acting as a prompt for parents to also attend for their own dental check-up. All FiCTION parents reported that their own dental health status was very important to them. FiCTION parents reported attending to be a positive role model to their child:

> "Yes. Well it's more the fact that I want them to see that I'm doing it so they should do it as soon as they get older they still….For me, probably not quite as frequently needs but I do it for them. For their benefit." (Parent 5, FiCTION, Female, 32 years old)

In contrast, some Non-FiCTION parents did not rate their dental health as anything other than "normal" and did not give it any increased consideration within their lives. When analysing this further, risk tolerance was only noted within the Non-FiCTION parents when justifying irregular dental attendance for themselves:

> "It's important – you know, even though I haven't been to the dentist, it's important to me and if I got – if I had a problem, I would get it sorted and I'd be – probably if I did have a problem, I would probably then become much more, you'd

*go much more regularly and start to – I almost need to actually have a problem to make me feel that I personally need to make more of an effort and there's maybe how I feel about it. At the same time, I'm, kind of, I'm pleased that my teeth – I've never had a filling ever, for example. I think I must have looked after them when I was younger and I can't be doing anything too wrong but I think I'd change if I actually had a problem." (Parent 11, Non-FiCTION (Ineligible), Male, 31 years old)*

Parents either approached their dental attendance as a matter of course, i.e., they went regularly (presumably at the interval advised by their dental team) because they felt it was "normal" that they should, or they had become a little erratic in attendance due to complacency (likely due to a history of good dental health):

*"Still probably as and when. I think, the last time I was there was about two years ago. It was like, I always think I'll make an appointment, I'll go for a check-up and I never kind of do it… "(Parent 7, Non-FiCTION (Ineligible), Female, 34 years old)*

*"I think…vital…I think every six months. And if you've got problems, then, you need to be able to step that to three months to make sure that there's no regression." (Parent 16, FiCTION, Male, 43 years old)*

**Theme 2: Poor dental health impacts on nutritional, psychological and social performance but dental practice selection and dental attendance have nothing to do with research.** More negative childhood experiences were reported by parents who had attended a dentist irregularly in childhood, irrespective of their current attendance pattern. Only two parents considered their childhood experiences positively; and both had limited dental treatment as a child. Interestingly both were male and neither's child was involved in the FiCTION RCT (due to ineligibility):

*"I preferred going to the dentist than getting my hair cut. That tells you everything. The dentist was fine. The pain was fine." (Parent 16, Non-FiCTION (Ineligible), Male, 43 years old)*

All parents reported that their child went to their dental check-ups regularly and the reasons given were the same for FiCTION and Non-FiCTION parents. The criteria that parents currently and historically used to select a suitable dental practice for themselves, and their family were similar between both groups of parents:

*"It (research) wouldn't actually concern me. I would actually just look for the dentist that was locality wise and best offered, the best healthcare for them." (Parent 5, FiCTION, Female, 32 years old)*

*"Probably again the locality. Um, I think, as well, the number of dentists within the practice because sometimes getting an appointment can be quite hard if you're… if there's an emergency of anything like that. And I think, for me as well, I tend to kind of ask family and friends and ask for… like, by reputation and things like that and how good they are." (Parent 7, Non-FiCTION (Ineligible), Female, 34 years old)*

## Participation in research

This section discusses the parents' journey from being introduced to the FiCTION RCT, through to joining the qualitative study and the effect of participating in research on their everyday lives. Analysis of the parent's views, knowledge and experiences, and perceptions revolved around four major themes and associated subthemes:

1) Research needs to be clearly justified.

2) Parents do not always have complete knowledge or understanding of a research study in which they are participating.

3) Parents will engage with further research if it is timely and relevant.

4) Research engagement can be challenging particularly for children and ethnic minorities.

**Theme 1: Research needs to be clearly justified.** Parents' views about the impact of their participation in research, via the dental practice, was the same between both groups of parents. There were, however, contrasting views between FiCTION and Non-FiCTION parents with respect to their thoughts about their dental practices being involved in research:

*"I think if the dental practice have a research practice, I think they are better from the other. Could be they give me a better service." (Parent 1, FiCTION, Male, age not reported)*

*"Yeah, it won't affect me. Yeah I won't change my mind based on whether they are doing research or not…." (Parent 3, Non-FiCTION (Ineligible), male, 41 years old)*

Most FiCTION parents thought that dental practices were participating in FiCTION either as a regulatory requirement for members of the dental team or to increase knowledge around the FiCTION topic, to inform local or national practice:

*"I think they probably have a legal requirement to do so. They've got a duty of care to look after their patients and look after them well." (Parent 14, FiCTION, Male, age not reported)*

Non-FiCTION parents were more likely to perceive that participation in the FiCTION RCT was to make the practice more attractive to patients.

**Theme 2: Parents do not always have complete knowledge or understanding of the study in which they are participating.** This theme covered parents' recollection of being approached to participate in the FiCTION RCT and their understanding of research concepts, such as random allocation and potential withdrawal from the research study. The screening process for identification of children for the FiCTION RCT was through a routine dental examination ('check-up'). As expected, more parents whose children were participating in FiCTION had an adequate understanding of the screening process than those whose children were not participating in FiCTION; however, there were no contrasting views noted between the Non-FiCTION sub-groups. In addition, most FiCTION parents remembered receiving written information whilst most Non-FiCTION parents did not. Given that the patient information leaflets were given before the child's screening appointment, it might be assumed that both groups should have had the same level of understanding. However, FiCTION participants may have read the patient information leaflet more thoroughly after the screening appointment leading to better understanding. Some practices also used notes-based screening to identify potentially eligible patients for the FiCTION RCT rather than by approaching parents first by post; it is unclear whether this would have impacted on their understanding of either the FiCTION RCT or this current study. All parents, except one, felt it was either "normal" to have been asked to participate in the FiCTION RCT, or reported neutral feelings about being asked. The parent who was initially suspicious about the recruitment process, still enrolled their child in the FiCTION RCT:

*"I was asking the question, how many kids who come to this practice will be taking part. So, I was wondering I heard things like random selection. I kind of think, well, actually, I know how that works. It's not always random, you know? That's me being a bit cynical. So, that was a question that came to my mind." (Parent 13, FiCTION, Male, 51 years old)*

More FiCTION parents reported neutral feelings about research participation than Non-FiCTION parents. Rather unexpectedly, Non-FiCTION parents were slightly more inclined to see it as "normal" to have been asked. There were no contrasting views between parents in terms of their rationale for participating in this qualitative study. The decision processes to take part in FiCTION varied for parents whose children were eligible to participate. A few parents felt that the decision to participate should lie solely with the clinician whilst others felt that the decision to participate, or not, was the child's. Only

one parent reported that they felt it had been a joint decision between parent and clinician. One parent felt the decision regarding participation had already been made by the clinician:

*"It wasn't as if there was an option for him to be part of it. It was, he is part of this and that's it type of thing. So, it was a case of right, okay then." (Parent 8, FiCTION, Female, 42 years old)*

A diversity of understanding regarding the term "RCT", none of which were particularly accurate, was noticed with both FiCTION and Non-FiCTION parents, although more FiCTION parents were, perhaps unsurprisingly, comfortable with the concept of random allocation. However, interestingly one FiCTION parent was unhappy with the concept of random allocation and thought he should choose the arm his child was in:

*"No, no I should decide. I will not let the computer choose. I should choose." (Parent 1, FiCTION, Male, age not reported)*

This highlighted the importance of parents understanding the concept of random allocation before a child is enrolled into an RCT. Non-FiCTION parents, whose child(ren) were not participating in the FICTION RCT due to ineligibility, gave a full range of opinions on random allocation from very favourable to very negative comments. There were no contrasting views between FiCTION parents and Non-FiCTION parents in terms of who would make the final decision to participate in FiCTION; parent, child, clinician or a combination of these stakeholders. What was noticeable, however, was that FiCTION parents had clearly experienced the journey through the decision-making process, whereas Non-FiCTION parents, due to ineligibility, were only considering the decision in the abstract. The parent whose child was eligible to participate in the FICTION RCT but had declined consent, was unhappy with random allocation and felt that the dentist should make the final decision:

*"Well I suppose if you're not happy with it then you would have to…. Either talk to your dentist or if there is nothing he can do then I suppose you would… if it effects these, me I'd move dentist. My kids come first so it's as simple as that." (Parent 4, Non-FiCTION (Eligible but Declined), Female, 47 years old)*

Since their child was still attending the same dental practice, this suggests they had felt empowered to make that decision without any negative consequences. There were also contrasting views between FiCTION and Non-FiCTION parents when considering the process of withdrawal from the RCT. FiCTION parents were more confident that withdrawing from the RCT completely or deviation from clinical protocol was acceptable, in comparison to Non-FiCTION parents. This may be due to increased discussion between the FiCTION parents and dental teams:

*"You can change your mind and opt out of it at any point. She always reminds you about that if, um, if (name) struggled." (Parent 6, FiCTION, Female, 34 years old)*

Since giving consent is an on-going active communication process throughout a clinical trial, it was expected that the FiCTION parents would remain thoroughly informed throughout. Some parents reported trying to take their child's thoughts into account when considering whether to become involved:

*"I think we would probably override any decision because obviously we are adult. But I think, if I knew they definitely weren't happy with it, I wouldn't push them". (Parent 7, Non-FiCTION (Ineligible), Female, 34 years old)*

**Themes 3: Parents will engage with further research if it is timely and relevant.** The majority of parents had not participated in research before being approached for the FiCTION RCT. This was reported to be largely due to a lack of opportunity rather than any reluctance on their behalf:

 

*"No, I don't think so…. Never been approached." (Parent 18, Non-FiCTION (Ineligible), Male, 56 years old)*

In addition, the majority of parents reported participating in the current qualitative study to increase their knowledge base or because of their previous experience with FiCTION:

*"To find out a bigger picture." (Parent 8, FiCTION, Female, 42 years old)*

*"You know, I've done one, I might as well do the other." (Parent 5, FiCTION, Female, 32 years old)*

Those parents with prior research experience reported their previous experiences positively which may have facilitated their participation in the FiCTION RCT:

*"She was coming out and asking him a series of questions and that's, uh, we took part in that survey and I, that was, that was, that was three year ago now like. But, ah, I can't remember the questions we're asked, we were answering to be honest with you but, ah, I do know that she found it very beneficial because we got a nice thank you letter at the end of it." (Parent 14, FiCTION, Male, age not reported).*

Parents were asked, based on their experiences with the qualitative study and the FiCTION RCT, whether they would be willing to take part in subsequent dental research studies. No participants reported that their experiences would stop them participating in further research; the factors influencing that remained the nature of the research and its relevance to them. However, a few participants reported that they may delay participating in further research studies as they felt they had contributed fully to the current study:

*"I would feel more inclined in a sense that I know what it's about and if it was a similar format, I think, if I could fit it in, but if another one came next week, I'd think, oh I've just done one, so I'm not…" (Parent 11, Non-FiCTION (Ineligible), Male, 31 years old)*

**Theme 4: Research engagement can be challenging particularly for children and ethnic minorities.** Both FiCTION and Non-FiCTION parents showed a range of opinions regarding their views on the public's willingness and ability to participate in research. As all parents had allowed their children to be screened for the FiCTION RCT, it was perhaps unsurprising that all parents believed that children should be involved in research. It was generally perceived that the quality and quantity of data gathered would be greater if children were involved:

*"You'll get completely different answers" (Parent 4, Non-FiCTION (Eligible but Declined), Female, 47 years old)*

FiCTION and Non-FiCTION parents' willingness for young children to participate in research was mixed. FiCTION parents were largely unconcerned about the age of the child whilst Non-FiCTION parents were a little bit more guarded in their willingness for young children to participate in research:

*"I think yeah, age matters." It matters… at least they should understand what he said". (Parent 17, FiCTION, Male, age not reported)*

The FiCTION RCT did not come with additional dental visits. However, the concept of children having to miss school to participate in a research study was a regular talking point. Parents' views varied greatly, from concerns that missing school was not a viable option, *""Definitely not" (Parent 7, Non-FiCTION (Ineligible), Female, 34 years old)* to acceptance that missing school for a short time was acceptable. Most parents felt that missing school was not ideal but that they may

be willing to make some concessions, depending on the circumstances. As part of the face-to-face interviews, parents were asked about their personal preferences regarding their preferred route of being approached to participate in research studies. A small number of parents reported a preference for online contact, due to ease of reply, security of the data set and cost effectiveness. Overwhelmingly, however, parents preferred to be asked by post because either it was more professional, less likely to be forgotten or more inclusive:

*"Post is fine. I really don't like the way this e-mail c\*\*\*'s going because if you haven't got a printer, you're knackered…. Um, I think this digital technology only's great, you know what I mean, if, if you've got access to good, good equipment." (Parent 14, FiCTION, Male, age not reported)*

In the current study, a number of parents from ethnic minorities were interviewed; on the whole, they felt ethnicity was not a barrier in research participation. However, one ethnic minority parent believed that their under-representation in research may be an issue. They elaborated that it was easy for ethnic minorities to be misunderstood both in terms of how they respond to questions and their interaction with the researcher:

*"In respect to ethnic minorities, I've worked with quite a lot of them as well. Em. Translation is massive. And, the thing with ethnic minorities is that their cultures are different. It's easy to presume em the way, say for example, a culture doesn't brush their teeth but they will chew cinnamon sticks. Well, that's their way of brushing their teeth, that's our version of it. But it's very easy eh to misunderstand with em the communication, very very easy." (Participant 10, Non-FiCTION (Ineligible), Female, 44 years old)*

The same parent felt that it was necessary for different recruitment techniques to be utilised to overcome these misinterpretation barriers, especially for first or second generation immigrants:

*"Explain the reason for the question. Make sure they or the translator understand the reason for the question and they will give you an answer, it might not be that question, but it's what you are looking for…The other thing is with foreign cultures, is especially in the West, people from different lands feel that the west look down on them. And they've almost all got: you think you're better than us. So there must be an awareness of that, as an interviewer (Parent 10, Non-FiCTION (Ineligible), Female, 44 years old)*

The parents interviewed in the qualitative study came from a range of educational backgrounds. There did not appear to be a relationship between willingness to participate in the FiCTION RCT and educational background. Nor did educational level seem to be related to parents' ability to define a RCT or the concept of random allocation of participants.

## Discussion

As described in the introduction, the objectives of this qualitative study were to investigate parents views, knowledge and experience regarding dental health and participating in research.

With respect to the first qualitative study objective, parents' views regarding their own dental health were varied. There were no noticeable contrasting views between FiCTION and Non-FiCTION parents' opinions of good dental health or their perception of the facilitators and barriers to achieving good dental health. All parents reported that their child went to their dental check-ups regularly and the reasons given were the same for FiCTION and Non-FiCTION parents. Considering that child attendance for dental care has previously been linked to a parent's attendance [35], it was a little surprising to find that not all participating parents reported that they themselves also attended the dentist regularly. This suggests that financial factors could be at play as whilst the children would have all been eligible for free NHS treatment, the parents may have had to pay for NHS dental care.

The interviews highlighted that, whilst the majority of the parents, irrespective of trial status, attended their dentist regularly, a minority were irregular attenders. Perhaps not unexpectedly, it was only some Non-FiCTION (Ineligible) parents who were irregular attenders, highlighting a potential contrast between FiCTION and Non-FiCTION (Ineligible) parents in terms of attendance patterns. This suggests that parents who were regular attenders themselves may be more likely to involve their child in research or, alternatively, that parents' motivation for attending regular dental check-ups may have increased because of their child's involvement in the FiCTION RCT. An unintended positive effect of dental research may be that it could act as a prompt for other family members to also attend for their dental appointments. Most parents had selected their current dentist by word of mouth and the convenience of the practice locality, with none having decided to attend based on the dental practice's research profile. Therefore, research involvement did not appear to be a major incentive to attend a particular practice.

With respect to the second qualitative study objective, most FiCTION parents understood the general aspects and advantages of participating in the trial, such as the nature of the study, the potential benefit to other children, the notion of voluntary participation and the possibility of withdrawal at any time. By contrast, a greater proportion of Non-FiCTION (Ineligible) parents struggled with these concepts, which might be expected in view of their much more limited exposure to the FiCTION RCT. Parents' views, knowledge and experience about participation in research were influenced by participation in the FiCTION RCT, but some of the more complex concepts around RCTs were not fully understood by either group. FiCTION parents' knowledge around the process for withdrawal from the trial was more extensive than that of Non-FiCTION (Ineligible) parents. This is likely due to their experiences within the FiCTION RCT, especially as dentists became more immersed in the RCT and became more familiar with trial processes themselves. This was likely to have a "knock-on" effect on the practice's FiCTION families becoming more aware and familiar with trial processes as the trial progressed. It is less clear whether Non-FiCTION (Eligible but Declined) parents had a more extensive knowledge than Non-FiCTION (Ineligible) parents as only one Non-FiCTION (Eligible but Declined) parent was included within the qualitative study. However, it was concerning that one of the FiCTION parents felt that the decision regarding participation was made for them. Given that the patient information leaflets were given before the child's screening appointment, it might be assumed that both groups would have had the same level of understanding. However, FiCTION parents or Non-FiCTION declined parents may have re-read the patient information leaflet more thoroughly after the screening appointment leading to better understanding. Both FiCTION and Non-FiCTION parents included some individuals who had struggled to explain the concept of random allocation to a trial arm and also seemed unclear about the need for random allocation to treatment rather than through clinician and/or patient choice.

The difficulty that participants had in describing randomisation, in expressing the purpose of the research or specific details about the FiCTION RCT's three arm design mirrors findings from previous medical RCTs involving children [36]. It also echoes the findings of a clinical trial involving parents of critically ill babies [37] where some parents gave seemingly appropriate descriptions of the trial but further examination highlighted areas of confusion. The parents in the present study also used or responded to terms such as "random" or "randomisation" as if they were familiar with them but further unpicking identified some uncertainty or incorrect interpretations of the terms.

It has been claimed that racial and ethnic minorities, especially in the USA, are less willing than non-minority individuals to participate in health research but these assumptions generally appear to be on the basis of analysis of single trial datasets [12]. Within the present qualitative study, ethnic minority participants largely felt ethnicity was not an issue for the associated FiCTION RCT. Reasons for perceived exclusion of minority ethnic groups are complex and previous medical research has reported that it is unclear whether the real issue is one of planned exclusion, inadvertent exclusion (e.g., through lack of study materials in minority languages), an explicit decision regarding non-participation or a mixture of these [38], all of which would result in under-representation. We know from previous medical research [39, 40] that engagement with communities and more personalised approaches are beneficial to increase the recruitment and

participation of patients from all communities, including minority ethnic groups, and the same approach is necessary for dental research studies.

It has previously been reported that, in general, those who attain a higher level of education have a more favourable view of medical research and are more aware of the approval processes [41]. From the current interviews, there did not appear to be a relationship between willingness to participate in the FiCTION RCT and educational background, however it was acknowledged that these parents had opted into the qualitative study and therefore may not be completely representative of the wider population.

Research conducted within primary dental care services remains limited [42]. Given that research participation is not common place or heavily advertised, unless the dental practice chooses to advertise their involvement, this is perhaps unsurprising. Considering that it is stated in the NHS Constitution that the NHS commits "to inform you of research studies in which you may be eligible to participate" [43] [p. 8] and that this should extend to dental patients as well, it will be interesting to see whether dental practices' research participation profiles enter the public domain in future. For the majority of dental practices involved in the FiCTION RCT, this was their first experience of undertaking research. There is growing support for all registered clinical trials to be required to publish their results [44]. If this does happen, it will be interesting to note whether research involvement does begin to become a factor used to attract potential patients to the practice, e.g., information included on the practice website/ posters displayed in their premises.

The NHS is in a 'critical condition' [45] and some politicians are openly calling for the NHS' founding principles to be abandoned and the taxpayer-funded, free at the point of need, health service to be scrapped [46]. Satisfaction with NHS dentistry has continued to collapse; as recently as 2019 this was at 60%, but it has now fallen to a record low of 20% [47]. Dissatisfaction levels (55%) are the highest for any specific NHS service asked about [47]. Despite the qualitative interviews suggesting that parents value the need for research in primary dental care, the mechanisms for this remain complex and challenging. Primary care dentistry is complex. There is a clear need for better quality evidence on which to base practice and one of the keys to generating this evidence is to conduct primary care dental research studies. The British Dental Association estimates 13 million adults – over 1 in 4 – are struggling to find NHS dental care [48]. Worse, evidence shows that even amongst those who do get appointments, it is patients with the greatest clinical need who miss out [46]. There is a recognised need to encourage patient involvement in research, with a focus on underrepresented groups such as young people, Black people and people of South Asian heritage [46]. This paper highlights that further research is required to explore the best methods to achieve engagement with patients in primary dental care research.

## Strengths and weakness of the study

The semi-structured interview design enabled potentially sensitive subjects to be explored delicately. By conducting the interviews in-person, the lead author (HC) was able to empower parents by putting them at ease before beginning the interview and by explaining that it was their experiences that were of interest. It was important to consider the best way to maintain the balance of power between the parent and interviewer, since by being responsible for introducing the topics and guiding the interview, HC could be seen as 'having the upper hand'. It was emphasised that it was the parent who had encountered the personal experience and thus was the "expert". In-person interviews were logistically more challenging, in terms of travelling, and may have been more of a burden on parents than a telephone interview would have been.

Parents were selected using purposive, maximum variation, sampling which allowed the interviewing of parents from different social and ethnic backgrounds and both mothers and fathers in Scotland and North-East England.

Interviews were completed when further interviewing generated no additional themes [20]. However, the concept of data saturation is widely contested and further interviews may have generated additional/different findings. In addition, including parents from the other three centres in the UK-based FiCTION study could have reached different assumptions. Although it would have been desirable to include additional younger parents who had completed the questionnaire survey to increase the age range, this option was limited since the vast majority of those parents were in their 30s or 40s.

Likewise, it would have also been advantageous to interview more parents who were eligible to participate in the FiCTION RCT but had declined, but this option was restricted by the very small number of parents that fitted this category.

An avenue not explored in any great detail with parents was, who primarily took their child to dental appointments. Most parents mentioned. within the interview. that they took the child to the dentist for check-ups, for practical reasons, but it was not possible to ascertain whether the FiCTION parents took their children to the treatment sessions. If parents were reliant on other guardians to take their child to the dentist after the initial recruitment. which required parental attendance, this could account for the poorer grasp of knowledge and understanding seen around some of the research aspects included in the qualitative interview.

Although the sample did include parents with a wide range of socio-economic characteristics, all with some FiCTION experience, only those who returned a completed baseline questionnaire were interviewed. No claims can be made about parents who would have been willing to participate in the qualitative study but did not engage in the quantitative question-naire study [16]. Whilst the data were collected some time ago (2015), satisfaction with NHS dentistry has only worsened [47]. Indeed the relevance and applicability of the study is even greater since the recognition of the importance of the incorporation of under-represented groups such as young people in research has grown over this time period.

## Conclusion

This study identified positive parental experiences and reported parents were happy to be involved in the FiCTION RCT if it had minimal impact on their child and would lead to improved treatment for future children. Parents were less concerned about knowing which arm their child was recruited to in the RCT as long as they could change their mind about being involved. Parents felt that the attitudes and motivations of the dentists themselves had influenced them to participate in the qualitative study. These findings should be particularly interesting for policymakers given the dearth of published pri-mary dental care research studies involving children.

There is a growing recognition that dental practices in the primary care NHS sector provide an excellent and relevant environment in which to carry out clinical dental research and provide an opportunity for all members of the dental team to develop and expand their roles into the research field. In addition to evaluating treatment outcomes, research to under-stand the practicality, feasibility, acceptability, expense and cost-effectiveness of new treatment regimens are crucial to its overall deliverability in NHS dentistry [49]. The present study confirms that, with regard to the conduct of research trials in primary dental care, research dedicated to identifying the best methods to achieve engagement with patients as potential participants is lacking and clearly needed [50].

## Supporting information

**S1 File. Supplement: Qualitative interview topic guide.**
(DOCX)

## Acknowledgments

We are grateful to the parents, giving so generously their time and also sharing their experiences with us. We are also grateful to the GDPs and their clinical and administrative teams who supported this study.

## Author contributions

**Conceptualization:** Heather Coventry, Anne Maguire, Catherine Haighton.

**Data curation:** Heather Coventry, Catherine Haighton.

**Formal analysis:** Heather Coventry, Anne Maguire, Elaine McColl.

**Funding acquisition:** Heather Coventry, Anne Maguire.

**Investigation:** Heather Coventry, Anne Maguire, Catherine Haighton.

**Methodology:** Heather Coventry, Anne Maguire, Catherine Haighton.

**Project administration:** Heather Coventry, Anne Maguire, Elaine McColl, Catherine Haighton.

**Supervision:** Anne Maguire, Elaine McColl.

**Writing – original draft:** Heather Coventry, Anne Maguire, Elaine McColl, Catherine Haighton.

**Writing – review & editing:** Heather Coventry, Anne Maguire, Elaine McColl, Catherine Haighton.

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
