## [Decision Letter · Decision Letter 0]

10 Nov 2025

Dear Dr. Coventry,

Thank you for submitting your manuscript to PLOS ONE. After careful consideration, we feel that it has merit but does not fully meet PLOS ONE’s publication criteria as it currently stands. Therefore, we invite you to submit a revised version of the manuscript that addresses the points raised during the review process.

We look forward to receiving your revised manuscript.

Kind regards,

Saima Aleem

Academic Editor

PLOS ONE

Journal Requirements:

2. Thank you for stating the following in your manuscript:

[This research was funded by the Centre for Oral Health Research at Newcastle University and NHS Education for Scotland (through the Scottish Dental Practice Based Research Network).]

[This research was funded by the Centre for Oral Health Research at Newcastle University and NHS Education for Scotland (through the Scottish Dental Practice Based Research Network).

The funders did not play any role in the study design, data collection and analysis, decision to publish, or preparation of the manuscript.]

Reviewers' comments:

Reviewer's Responses to Questions

**Comments to the Author**

1. Is the manuscript technically sound, and do the data support the conclusions?

Reviewer #1: Yes

Reviewer #2: Yes

2. Has the statistical analysis been performed appropriately and rigorously?

Reviewer #1: N/A

Reviewer #2: N/A

3. Have the authors made all data underlying the findings in their manuscript fully available?

Reviewer #1: Yes

Reviewer #2: No

4. Is the manuscript presented in an intelligible fashion and written in standard English?

Reviewer #1: Yes

Reviewer #2: Yes

Reviewer #1: Abstract

I recommend that the authors spell out “NHS” in full at its first mention before using the abbreviation to ensure clarity for all readers.

Introduction

I recommend that the authors spell out “NIHR” in full at its first mention before using the abbreviation to ensure clarity for all readers.

The introduction would benefit from stating potential barriers related to parents not fully understanding the importance of research, and specifically RCTs, in the context of pediatric dental care.

Methods

In the data analysis section, the presentation of multiple analytic approaches (essentialist, contextualist, constructionist thematic analysis, framework method) makes the process difficult to follow. It would be helpful for the authors to clarify which approach was primarily adopted, and how the different methods were integrated or distinguished.

It would be useful to include a participant flow chart for the two study locations. For example, indicating the number of eligible parents at each site, the number who completed the baseline questionnaire, the number of parents contacted, and the number of parents interviewed. This addition would enhance transparency regarding recruitment across sites.

Results

In the participant profile table, it would be helpful to provide further detail on the category “other” for parent ethnicity. Clarifying what this category includes would improve transparency and interpretability of the data.

In the description accompanying the participant profile table, I suggest including the mean age of the parents.

The Results section is at times difficult to follow. For example, under “Dental Health” themes 2 and 3 currently contain relatively little information. It may strengthen the presentation to consider reorganizing or merging these themes in a way that avoids sections with limited content.

Discussion

The discussion section should be made more concise by focusing on the key takeaways that directly relate to the study’s objectives. Emphasizing how these findings connect to the current literature would strengthen the interpretive value and clarity of the manuscript.

Strengths and weakness of the study

Line 669 to 672: The text in this section appears to be incomplete, with some words or parts of sentences missing. Please review and revise for clarity.

The age of the data set (2015) may affect the relevance of the findings. Consider discussing any changes in context since 2015 that might influence interpretation or applicability.

Reviewer #2: This study provides insight into clinical trial recruitment and patient engagement. The methodology along with the findings section provide strong evidence for the validity of the conclusion.

Two main reasons for recommending Minor Revisions:

1- Data Availability: Since this is a qualitative study, full data sharing is not recommended due to participant confidentiality and ethical considerations. It will be helpful to clarify this in the Data Availability Statement. A possible way to phrase it is as follows:

[Qualitative interview transcripts may contain personally identifiable information that cannot be shared with the public. Researchers who meet the criteria for accessing confidential data can access it through the Newcastle University Research Data Service].

2- Rigor: it is important to outline in the Results or Discussion section when saturation was reached. This minor addition will ensure full adherence to qualitative reporting standards [e.g., saturation was reached after the 15th/18th interview].

**Do you want your identity to be public for this peer review?** For information about this choice, including consent withdrawal, please see our Privacy Policy

Reviewer #1: No

Reviewer #2: No

---

## [Author Response · Author response to Decision Letter 1]

15 Jan 2026

Please find enclosed revised manuscript.

Funding information has been removed from the manuscript. The Funding Statement does not require revision. Please change the online submission form on my behalf.

---

## [Editor Report · Decision Letter 1]

19 Feb 2026

Parental attitudes to randomised controlled trials in primary dental care: A qualitative study

PONE-D-25-39208R1

Dear Dr. Coventry,

We’re pleased to inform you that your manuscript has been judged scientifically suitable for publication and will be formally accepted for publication once it meets all outstanding technical requirements.

Kind regards,

Saima Aleem

Academic Editor

PLOS One
---

## [Editor Report · Acceptance letter]

PONE-D-25-39208R1

PLOS One

Dear Dr. Coventry,

I'm pleased to inform you that your manuscript has been deemed suitable for publication in PLOS One. Congratulations! Your manuscript is now being handed over to our production team.

Kind regards,

on behalf of

Dr. Saima Aleem

Academic Editor

PLOS One